# The Investigation of the Antitumor Agent Toxicity and Capsaicin Effect on the Electron Transport Chain Enzymes, Catalase Activities and Lipid Peroxidation Levels in Lung, Heart and Brain Tissues of Rats

**DOI:** 10.3390/molecules23123267

**Published:** 2018-12-10

**Authors:** Gizem Kursunluoglu, Dilek Taskiran, Hulya Ayar Kayali

**Affiliations:** 1Izmir Biomedicine and Genome Center (IBG), İzmir 35340, Turkey; gizemkursunluoglu@ibg.edu.tr; 2Department of Chemistry, The Graduate School of Natural and Applied Sciences, Dokuz Eylul University, İzmir 35160, Turkey; 3Department of Physiology, Ege University School of Medicine, İzmir 35100, Turkey; dilek.taskiran@ege.edu.tr; 4Department of Chemistry, Division of Biochemistry, Faculty of Science, Dokuz Eylul University, İzmir 35160, Turkey

**Keywords:** cisplatin, capsaicin, ETC, CAT, LPO, ATP–ADP

## Abstract

Cisplatin is one of the most active cytotoxic agents in cancer treatment. To clarify the interaction with mitochondria, we hypothesize that the activities of mitochondrial electron transport chain (ETC) enzymes succinate dehydrogenase (SDH) and cytochrome c oxidase (COX), nucleotide levels, as well as levels of catalase (CAT) enzyme and membrane lipid peroxidation (LPO) can be affected by cisplatin. There was a significant decrease of both SDH and COX activities in the lung, heart, and brain tissues at the 1st day after cisplatin exposure, and the observed decreased levels of adenosine triphosphate (ATP) and adenosine diphosphate (ADP) in comparison with the control could be because of cisplatin-induced mitochondrial dysfunction. The investigations suggested that cisplatin inhibits SDH, COX, and ATP synthase. The higher LPO level in the studied tissues after 1 and 4 days post-exposure to cisplatin compared to control can be inferred to be a result of elevated electron leakage from the ETC, and reactive oxygen species (ROS) can lead to wide-ranging tissue damage such as membrane lipid damage. Consequently, it was observed that capsaicin may have a possible protective effect on ETC impairment caused by cisplatin. The activities of SDH and COX were higher in heart and brain exposed to cisplatin + capsaicin compared to cisplatin groups, while LPO levels were lower. The investigated results in the cisplatin + capsaicin groups suggested that the antioxidant capacity of capsaicin scavenges ROS and prevents membrane destruction.

## 1. Introduction

Mitochondria are the energy factories of the cell, and this energy is used in the electron transport chain to pump protons across the inner mitochondrial membrane from the inner matrix to the intermembrane space, and a strong hydrogen concentration gradient is produced [1]. There are four large proteins associated with the mitochondrial electron transport chain (ETC) called complexes I–IV, bound to the inner membrane of the mitochondria, and electrons and protons are passed through electron carriers. At the end of these steps, the electrochemical gradient generates adenosine triphosphate (ATP) via the phosphorylation of adenosine diphosphate (ADP) to ATP.

Mitochondrial oxygen metabolism is the dominant source of O_2_^−^ that results from incomplete coupling of electrons and H^+^ with oxygen in the electron transport chain [2]. A free radical is any species capable of independent existence that contains one or more unpaired electrons, and the unpaired electrons of oxygen react to form partially reduced highly reactive species that are classified as reactive oxygen species (ROS), including superoxide (O_2_^−^), peroxyl radical, hydrogen peroxide (H_2_O_2_), and hydroxyl radical [3,4]. Several drugs, toxins, and agents that are used extensively in life induce electron leak in the mitochondria. One such agent is cisplatin, a chemotherapeutic agent widely used for the treatment of several types of cancers (e.g., testicular, ovarian, head and neck, cervical, colon, lung, and brain). Cisplatin is a major anticancer drug and a remarkable crosslinking agent that reacts indirectly with nitrogen atoms on DNA and kills cancer cells by damaging their DNA. Cisplatin’s antineoplastic effect depends on DNA cross-link formation [5].

Mitochondrial DNA is a particularly vulnerable target because of its proximity to the electron transport chain constituents [6]. The mitochondrial electron transport chain contains several redox centers that may leak electrons to oxygen, constituting the primary source of O_2_^−^ in most tissues, and this electron leak is associated with cisplatin-induced toxicity [7]. Therefore, cisplatin has several toxicities, and there are some cellular events observed after cisplatin-induced toxicity, such as decrease in protein synthesis, induced membrane lipid peroxidation (LPO), mitochondrial dysfunction, and DNA injury. This is a result of free radical generation and the insufficiency of molecules like superoxide dismutase (SOD), catalase (CAT), and glutathione peroxidase (GPx) [8,9,10]. These assertions are supported by a variety of studies, including those that demonstrate a protective role of free radical scavengers such as vitamin E and glutathione in cisplatin-mediated cytotoxicity [11,12,13,14,15,16]. Antioxidants serve to keep down the levels of free radicals, permitting them to perform useful biological functions without excessive damage [3]. Capsaicin is the major capsaicinoid present in peppers, and has been reported to exert analgesic, antimicrobial, anti-inflammatory, and antioxidant properties [17]. Its antioxidant effect is investigated in this study. This study provides the basis for understanding cisplatin’s toxicity and the protective effects of capsaicin on mitochondrial ETC enzymes such as succinate dehydrogenase (SDH) and cytochrome c oxidase (COX), as well as their effects on the antioxidant CAT enzyme and membrane damage indicator LPO in lung, heart, and brain tissues of the rat. 

## 2. Materials and Methods

### 2.1. Reagents

Cis-diammineplatinum(II) dichloride (cisplatin), capsaicin, trichloroacetic acid (TCA), and ascorbic acid were obtained from Sigma (Sigma-Aldrich GmbH, Sternheim, Germany), and thiobarbituric acid (TBA), and H_2_O_2_ were from Merck (Darmstadt, Germany). Other reagents used in experiments were of analytical grade and obtained from Sigma. Cisplatin was dissolved in saline solution; capsaicin was dissolved in ethanol.

### 2.2. Animals

Male Sprague Dawley adult rats (12 weeks, 450–500 g) were maintained in the laboratory in a temperature-controlled room on a 12-h light, 12-h darkness schedule, and rats were fed commercial rat chow and water ad libitum. Male Sprague Dawley adult rats were randomly divided into three groups (*n* = 6): control, cisplatin, and cisplatin + capsaicin group.

The animals were handled under the prescriptions for animal care and experimentation of the pertinent European Communities Council Directive (86/609/EEC), and all the procedures were approved by the Institutional Animal Ethics Committee of Ege University.

### 2.3. Cisplatin and Capsaicin Treatment

A single dose of cisplatin (5 mg/kg body weight) was administered intraperitoneally (i.p.) in rats, and they were sacrificed 1, 4, 7, and 14 days after of cisplatin treatment. Then, the lung, heart, and brain were collected immediately and used for biochemical studies. The control group rats received saline on the 1st day [18,19].

Capsaicin powder was dissolved in ethanol to reach a final concentration of ethanol 0.625% in serum. Cisplatin (5 mg/kg body weight) was injected in a single dose and after 7 days capsaicin (10 mg/kg body weight) was injected intramuscularly. The initial capsaicin injection was carried out just after cisplatin injection [19]. Capsaicin dose was chosen based on the previous literature examining the protective effect of capsaicin against drug toxicity. For control groups, physiological serum was injected in rats on the 1st day.

### 2.4. Cisplatin Levels

The lung, heart, and brain tissues were placed at 105 °C for approximately 24 h until the weight remained constant. Then, 9 mL HCl and 3 mL HNO_3_ were directly added onto the dry tissues, and then this remained in a microwave oven for UV decomposition for about an hour. Their volumes were completed to 20 mL with distilled water. The samples were filtrated with black band-type filter papers. The terminal half-life of cisplatin is 58–73 h. Therefore, the 1st and 4th days of all studied tissues were analyzed for cisplatin levels with inductively coupled plasma mass spectrometry (ICP/MS).

### 2.5. Isolation of Mitochondria

Lung, heart, and brain were washed in cold (4 °C) saline solution and cut into pieces, then minced and homogenized. The minced tissues were re-suspended in isolation buffer (1:15 *w*/*v*) containing 5 mM HEPES, pH 7.4, containing 1 mM EDTA and 300 mM sucrose. Sample tissues were homogenized at 8000 and 9500 rpm for different amounts of time based on tissue types in the ice. The heart tissue was homogenized at 9500 rpm for 50 s, and the lung and brain tissues were homogenized at 8000 rpm for 50 and 20 s, respectively. Tissue suspensions were ground in 1.5 mL plastic vials and centrifuged at 2000 rpm for 15 min, and cell debris was removed. The supernatant was centrifuged at 12,000 rpm for 15 min. Final pellets contained mitochondria. Before assaying, the mitochondrial pellets were re-suspended in isolation buffer and used for succinate dehydrogenase and cytochrome c oxidase activity assay.

### 2.6. Succinate Dehydrogenase Activity Assay

Succinate dehydrogenase in the mitochondrial pellet was assayed by measuring the initial rate of decrease in dichloroindophenol (DCIP) absorbance at 600 nm. The reaction mixture contained 50 mM potassium phosphate buffer, pH 7.0, 1.0 mM EDTA, 20 mM sodium succinate, 3 mM sodium azide, 5 μL enzyme solution, and 32 µM DCIP [20].

### 2.7. Cytochrome c Oxidase Activity Assay

COX in the mitochondrial pellet was assayed by measuring the initial rate of decrease in cytochrome c absorbance which was reduced by ascorbic acid at 550 nm. The reaction mixture contained 87.5 mM potassium phosphate buffer, pH 7.0, 30 μM reduced cytochrome c, and 50 μL enzyme solution. Cytochrome c in 10 mM potassium phosphate buffer, pH 7.0, was reduced by adding ascorbic acid and monitoring the absorbance at 550 and 565 nm. The blank solution included 90 mM potassium phosphate buffer, pH 7.0, 30 μM reduced cytochrome c, and potassium ferricyanide (K_3_[Fe(CN)_6_]) 2.5 mM. K_3_[Fe(CN)_6_] was included in the blank only in order to completely oxidize the reduced cytochrome c [21].

### 2.8. The Cytosolic Preparation for CAT Activity

Tissues homogenized in isolation buffer (1:15 *w*/*v*), lung, heart, and brain were homogenized at 8000 rpm for 90 s in ice. The sample suspension was ground in 1.5 mL plastic vials and centrifuged at 2000 rpm for 15 min, and cell debris was removed. The supernatant was used for the CAT assay. Catalase (CAT) activity in cytosol was determined in crude extract by the method of Aebi [22].

### 2.9. Lipid Peroxidation

Tissues were homogenized in isolation buffer (1:3 *w*/*v*), pH 7.5. Then, 500 µL homogenate was transferred into 2.5 mL 10% TCA, incubated 90 °C for 15 min, cooled, and then centrifuged at 3900 rpm for 10 min. Subsequently, 2 mL supernatant was added into 1 mL 0.675% TBA solution. The mixture was incubated 90 °C for 15 min. After cooling, the absorbance was measured 532 nm. Malondialdehyde (MDA), an end-product of fatty acid peroxidation, reacts with TBA and forms a colored complex. This complex has maximum absorbance at 532 nm. MDA values in nanomoles were calculated from the absorbance coefficient of MDA-TBA complex at 532 nm, 1.56 × 105 mol^−1^ × cm^−1^ [23].

### 2.10. Nucleotide Levels

The samples were prepared using the protocol of Cardoso et al. and Masubuchi et al. with some modification [24,25]. The minced tissues were homogenized, and then 1 M HClO_4_ (*w*/*v*) was transferred into the homogenates in a volume equal to five times their weights. They were centrifuged for 15 min at 5000 rpm. The supernatants were neutralized with 1 M K_2_CO_3_ and then centrifuged again. The clear supernatants were injected into HPLC for determining the levels of cytosolic adenine nucleotides.

A Thermo ODS-2 Hypersil column (250 × 4.6 mm) was used to obtain adenine nucleotide levels. The analysis was conducted under the following conditions: mobile phase, 50 mM aqueous triethylamine (TEA) buffer (adjusted with phosphoric acid to pH 6.5; A) and acetonitrile (B). Gradient elution was performed from 99A/1B in 10 min to 95A/5B and changed in another 10 min to 92.5A/7.5B. Each run was followed by a 5-min wash with 70B/30 parts 0.1% phosphoric acid. Detection wavelength, flow rate, and column temperature were set to 254 nm, 1 mL/min, and 20 °C, respectively [26].

### 2.11. Protein Determination

The protein content was determined by the method of Bradford [27]. Bovine serum albumin was used as standard.

### 2.12. Statistical Analysis

The results were expressed as mean ± SD of the number of experiments (*n*) indicated in the legend of the figures. ANOVA followed by post hoc Tukey’s test, one of the multiple comparisons, was used for statistical significance analyses. Additionally, comparisons between enzyme activities and different tissues were made with Pearson correlation at each cisplatin exposure day. The Shapiro-Wilk test was used to test the normality assumption. Analysis was performed using SPSS software (SPSS 15.0 Windows, US Government, 233 South Wacker Drive, 11th Floor Chicago).

## 3. Results

This study was primarily designed to determine whether or not cisplatin was taken up in the lung, heart, and brain tissues of cisplatin-exposed adult male Sprague Dawley rats. We evaluated the activities of ETC enzymes such as succinate dehydrogenase (SDH) and cytochrome c oxidase (COX), the levels of adenine nucleotides, the activity of the antioxidant enzyme CAT, and levels of LPO involved in membrane damage in all studied tissues. It was determined that there is a possible protective effect of capsaicin in lung, heart, and brain tissues against cisplatin toxicity. The studied results were compared with control group according to 1st, 4th, 7th and 14th days after cisplatin exposure.

### 3.1. Levels of Platin in Different Tissues of Male Sprague Dawley Adult Rats

Cisplatin which is a potent cancer drug enters the cells its chloride ligands are replaced by water, forming aquated species that react with nucleophilic sites in cellular macromolecules [28]. Uptake of the cisplatin was determined by ICP/MS in the lung, heart, and brain tissues on 1st and 4th days following a single dose of cisplatin exposure. The results indicated that cisplatin was transported to the lung, heart, and brain tissues of the rats. The levels of cisplatin reached a maximum at the 1st day in the brain and lung tissues, and these levels were respectively 0.24, 0.24, and 0.15 ppm cisplatin in 1 g of brain, lung, and heart tissue (Table 1). The lowest cisplatin level for the 1st day was found in the heart. The cisplatin level on the 4th day was almost zero for brain, while it decreased 3.85-fold for heart compared to the 1st day. However, the cisplatin levels in lung did not change markedly on the 4th day compared to the 1st day.

### 3.2. Variations in SDH Activities

Succinate dehydrogenase (SDH) or succinate-coenzyme Q reductase, an FAD-containing enzyme, is one of the important enzymes in the respiratory electron transfer chain, located in the mitochondrial inner membrane matrix. Figure 1 depicts the effects of cisplatin on the activities of SDH enzyme in all studied tissues. The SDH activity levels in the lung of the cisplatin group on day 1 post-exposure were 19.84 ± 3.96 IU/mg, significantly lower than control for all studied days, 38.80 ± 4.96, 60.00 ± 3.02, 62.04 ± 3.84, and 45.00 ± 5.01 IU/mg for days 1, 4, 7, and 14, respectively (*p* < 0.05). The SDH activity in the lung tissue on day 4 increased significantly compared to the 1st day (*p* < 0.05), but in lung it was not higher in comparison with control groups, in contrast to the brain tissue (Figure 1a,c).

The SDH activity in heart tissues of cisplatin-exposed rats on the 1st day post-exposure was 53.80 ± 4.15 IU/mg, markedly decreased compared to control, 95.00 ± 1.02 IU/mg (*p* < 0.05). However, the rate of SDH activity levels in heart of cisplatin groups compared to control decreased to be 0.9%, 6.7% at 7th, 14th days, respectively (Figure 1b). In the heart tissues of cisplatin groups, the SDH activity was significantly increased on days 1, 4, 7, and 14 post-exposure (*r* = 0.9). Figure 1c represents the effects of cisplatin on the SDH activity in brain tissues. The activity in the cisplatin group decreased on the 1st day, 46.17 ± 1.10 IU/mg, compared to control value, 85.79 ± 14.01 IU/mg. However, the SDH activity in the cisplatin group increased 3.5-fold on day 4, 160.80 ± 3.12 IU/mg, compared to the 1st day, 46.17 ± 1.10 IU/mg. Comparing the SDH activity levels in all control and experimental tissues, the lowest activity was determined in lung for all investigated days.

### 3.3. Variations in COX Activities

Cytochrome c oxidase is a multiprotein complex composed of subunits, and is known as the mitochondrial respiratory chain complex IV. COX is the terminal complex of the mitochondrial respiratory chain, and the site where over 90% of inhaled oxygen is consumed [29,30]. Figure 2a shows that the activity of COX in the lung of cisplatin groups showed similar tendency to the SDH activity. In the lung tissue of the cisplatin group, the COX activities were decreased to be 2.98 ± 0.21, 3.18 ± 0.49, 4.38 ± 0.35, and 2.85 ± 0.42 IU/mg in comparison with control levels 5.24 ± 0.59, 5.42 ± 0.33, 5.15 ± 0.29, and 5.07 ± 0.28 IU/mg on days 1, 4, 7, and 14 post-exposure, respectively.

The COX activity levels in the heart tissue reduced 1.20, 1.21, and 1.18-fold on the 1st, 4th, and 7th days post-exposure for cisplatin groups, respectively (Figure 2b). However, this activity of the cisplatin group was 2.04-fold higher (1.72 ± 0.05 IU/mg) compared to control (0.84 ± 0.18 IU/mg) on day 14 (*p* < 0.05). The activities of COX in the brain tissue were markedly reduced for the 1st day compared to control (*p* < 0.05), and the highest COX enzyme activity of all investigated tissues was observed for brain on the 7th day, as 3.2 ± 0.38 IU/mg (Figure 2c). Furthermore, positive correlations were obtained between COX activities and cisplatin exposure on the 1st, 4th, and 7th days in the lung (*r* = 0.92), heart (*r* = 0.94), and brain tissues (*r* = 0.96).

### 3.4. Variations in CAT Activities

Catalase is an important antioxidant enzyme that catalyzes the conversion of harmful H_2_O_2_ to water and molecular oxygen. In the first step, superoxide (O_2_^−^) generated by electron leakage in the ETC is dismutated to H_2_O_2_ by another antioxidant enzyme, superoxide dismutase [31,32]. In a second step, the produced H_2_O_2_ is reduced to H_2_O by CAT. Figure 3a shows that the CAT activity in the lung tissues of the cisplatin group (11.63 ± 2.12 IU/mg) was markedly lower than the control group, at 29.00 ± 1.17 IU/mg on the 1st day (*p* < 0.05). On the contrary, the CAT activity level in the lung tissues was higher in the cisplatin group on the 4th day, at 33.66 ± 1.95 IU/mg. The activities of CAT in the heart of the cisplatin group were significantly higher compared to control as 30%, 82%, and 51% for the 1st, 4th, and 7th days, respectively (*p* < 0.05) (Figure 3b). By the end of the 14th day after cisplatin injection, the CAT activity was reduced compared to control in heart tissues, and a negative correlation was obtained between the COX and CAT activity levels in the heart tissue of cisplatin groups for all observed days (*r* = −0.952). On the other hand, CAT activity in the brain was not determined due to low CAT activity in the peroxisome [33].

### 3.5. Variations in LPO levels

To obtain the lipid peroxidation levels, the malondialdehyde (MDA) concentrations were investigated in the lung, heart, and brain tissues. The LPO levels in all studied tissues after cisplatin treatment on the 1st and 4th days were markedly increased compared to controls (*p* < 0.05) and nearly reached the control levels for the last 7 days (Figure 4a–c). Furthermore, positive correlations were obtained between the lung and heart of cisplatin groups (*r* = 0.989).

### 3.6. Variations in Nucleotides Levels

A proton gradient is used to synthesize ATP and the production of electron flow. Figure 5a–c show that the ATP levels in the lung, heart, and brain tissues which were treated with cisplatin were markedly lower than in control groups on the 1st day (*p* < 0.05), which is a similar tendency to the reduction in the activities of SDH and COX. For the following days, the ATP levels in lung and brain tissues significantly rose and nearly reached the control levels. However, they were still lower for heart on 7th and 14th days compared to control.

The reduction in the ADP levels were 42.5%, 17.3%, 25%, and 55.5% in the lung tissues compared to control, and this reduction were 50%, 6.6%, 2.7%, and 63.4% in the heart tissues compared to control on the 1st, 4th, 7th and 14th days, respectively (Figure 6a,b). A positive correlation was obtained between lung and brain of cisplatin-exposed rats in the ATP levels during all periods (*r* = 0.956). However, the ADP levels markedly increased on the 1st day (*p* < 0.05). This level almost reached control levels on 4th, 7th, and 14th days in the brain tissue (Figure 6c).

### 3.7. The Enzyme Activities of SDH, COX, CAT, and LPO and Nucleotide Levels in Lung, Heart, and Brain of Rats Treated with Cisplatin and Capsaicin

The further question to be answered is regarding the protective effect of capsaicin on the enzyme activities of SDH, COX, and CAT, and levels of LPO and nucleotides in the lung, heart, and brain tissues. The SDH activities increased by 13% and 24% in the heart and brain tissues of rats treated with capsaicin and cisplatin compared to cisplatin groups, respectively. Furthermore, as shown in Figure 7a, the SDH activities markedly decreased by 41% in the lung of rats treated with cisplatin + capsaicin (*p* < 0.05). In the heart tissue of the cisplatin + capsaicin group, the COX activity was markedly higher than the control and cisplatin groups (*p* < 0.05) (Figure 7b). However, in the cisplatin + capsaicin group the COX activity decreased by 29.5%, 30.3% compared to control for lung and brain tissues, respectively (*p* < 0.05). As can be seen in Figure 7c, in the cisplatin + capsaicin groups, the CAT activities in the heart markedly decreased by 34%, while it significantly increased 71% for lung compared to cisplatin groups. The LPO levels reduced in the cisplatin + capsaicin groups for the lung, heart, and brain tissues compared to cisplatin groups 37.6%, 21.8% and 27.9%, respectively (Figure 7d). In contrast to ADP levels, the ATP levels markedly increased in the heart of cisplatin + capsaicin groups in comparison with cisplatin groups. On the other hand, for cisplatin + capsaicin groups ATP levels increased by 14% and 5% compared to control in the lung and brain, respectively (Figure 7e–f).

## 4. Discussion

Cisplatin is extensively used and plays a role in the treatment of different cancers [28]. Nephrotoxicity, neuropathy, ototoxicity, and gonad toxicity are clinically important dose-limiting side effects that limit the use of cisplatin. After cisplatin enters the cell, it exerts its cytotoxic effect in the cell leading to apoptosis, necrosis, oxidative stress, fibrogenesis, inflammation, hypoxia, and mitochondrial damage by activating a multi-signal transduction pathway. Mitochondria are the major target of cisplatin, and therefore the ETC is affected by cisplatin toxicity via the induction of ROS generation, electron leaks, and mitochondrial impairment.

In the current study, cisplatin transport was carried out to all studied tissues, and decreases in cisplatin levels were low in the brain compared to the lung and heart tissues on the 4th day. It is suggested that the blood–brain barrier protects against certain molecules. In other studies coherent with our results, cisplatin levels in the brain reduced significantly after cisplatin injection over the course of a few days. It may be explained that decreases in cisplatin level to protect certain molecules from reaching the central nervous system and in particular, it is impervious to hydrophilic substances such as cisplatin [34].

The cisplatin toxicity involved in SDH enzyme activity was evaluated in the lung, heart, and brain tissues. After cisplatin exposure in the lung, heart, and brain tissues on the 1st day, the significantly decreased SDH activity compared with control group could be due to cisplatin-induced mitochondrial dysfunction and the SDH enzyme was inhibited [26]. A number of apoproteins of the mitochondrial electron transport complexes I, III, IV, and V are encoded by mtDNA, but complex II is totally encoded by nuclear DNA [35] because cisplatin is a remarkable DNA-crosslinking agent. After cisplatin treatment, the lowest level of SDH activity was determined in the lung tissues compared with the brain and lung tissues. However, the SDH activities in all studied tissues increased and nearly reached their controls during 14 days following cisplatin exposure.

The marked increase in SDH activity for the 4th day was probably to protect the brain and decrease neurotoxic effect. In addition, after cisplatin exposure, the cisplatin levels reached a non-detectable level in the brain tissue. These SDH activities results were explained by suggesting that much more ATP is required for cisplatin elimination from the blood-brain barrier due to elevated SDH enzyme activity.

The findings of similar SDH enzyme activities on 7th and 14th days in the heart compared to control could be explained by rapid adaptation due to the lowest cisplatin levels in the heart compared to other studied tissues from the initial day of cisplatin-exposed rats.

COX is the one of the enzymes in electron transport chain and was investigated in the lung, heart, and brain tissues of cisplatin-treated rats. The activities of COX in all studied tissues were found to be markedly reduced in cisplatin groups compared with their controls. Like the report of Reference [36], in the heart tissue of mice exposed to 10 driamycin, membrane rigidity was accompanied by the inhibition of COX enzyme [37]. Moreover, in accordance with the study findings, significant differences were also observed between the three tissues. Gerschenson et al. reported that ETC enzymes encoded by mitochondrial DNA showed tissue-specific differences after cisplatin exposure [37].

There was no significant change in the COX activity between the 1st and 4th days in the lung of cisplatin-exposed rats because of the lowest decrease by 4.1% in cisplatin levels obtained in lung tissue (Table 1). The COX enzyme activity was highest in the lung tissue relative to the heart and brain, and previous literature reported that this activity in the brain was higher than heart tissues, as shown in our findings [38]. The COX activities in lung, heart, and brain tissues were found to be markedly decreased in cisplatin-exposed rats compared to control, as with the SDH activity. These observations strongly suggest that cisplatin may induce alterations in the activity of electron transport chain enzymes.

CAT activity is indicative of the efficient involvement of the enzymes in the removal of hydrogen peroxide, which in turn protects the organs from oxidative damage [39]. In the present study, the CAT activity was significantly reduced in the lung tissue of cisplatin-treated rats compared with the control groups on the 1st day post-exposure. This may reflect that cisplatin impaired the mitochondrial ETC of the lung. In addition, CAT activity on the 4th day was higher in comparison with the 1st day because cisplatin was not significantly excreted by the lung between the 1st and 4th days. Catalase overexpression is to protect the cell death against ROS such as H_2_O_2_ which is carried out after cisplatin toxicity [40]. The marked increase in the CAT activity did not show a parallel with the activities of SDH and COX in the heart tissues of cisplatin-treated rats. This may be explained by the fact that H_2_O_2_ was produced by other mechanisms besides the dismutation of O_2_^−^ which is produced by ETC enzymes. Nevertheless, in our study the CAT activity in the brain was not determined due to low activity of CAT in the peroxisome, and Uysal et al. also defined similar pieces of information in their study [33]. Besides, brain tissue is not particularly enriched in antioxidant enzymes such as catalase [30]. Other studies showed that CAT enzyme activity levels were highly active in the liver and kidney, in contrast to the specific activity of brain CAT, which was found to be lower. Jena et al. found that CAT activity was minimal in the brain tissue compared to other studied tissues [41].

We examined the changes in lipid peroxidation levels of lung, heart, and brain tissues of cisplatin-exposed rats to clarify the relationship between LPO and cisplatin-mediated toxicity. The negative correlation between the LPO levels and SDH and COX activities in the lung tissue of cisplatin-treated rats on the 1st and 4th days may suggest that augmented activity of ROS causes a consequent increase in MDA production and a decrease in enzymatic and non-enzymatic antioxidant systems [42,43]. The higher LPO level in the studied tissues after cisplatin exposure on the 1st and 4th days compared to control may be explained by the high electron leakage associated with SDH. Furthermore, ROS can potentially damage membranes by triggering lipid peroxidation. These results were interpreted as suggesting that the membrane damage increased on the 1st and 4th days in all investigated tissues of cisplatin-exposed rats, and these findings showed parallelism with the previous study [44,45].

The ATP levels markedly decreased compared to control groups and displayed similar tendency with the activities of COX and SDH on the 1st day, and these observations strongly suggested that cisplatin affects the electron transport chain, complexes II–IV–V, in all studied tissues. Nevertheless, the decreased ATP rate in the cisplatin-exposed tissues compared to control was arranged as lung, heart, and brain tissues at the 1st day post-exposure. Furthermore, the highest and lowest cisplatin levels at the 1st day were observed in lung and brain, respectively. The determination of a negative correlation between ATP and cisplatin levels in all three studied tissues may be explained by the inhibition of the ATPase enzyme by cisplatin, as shown in other research [46].

When compared to ATP levels in the studied tissues of cisplatin-treated groups on the 1st and 4th days, they did not markedly change in the lung in contrast to the increasing trends in heart and brain tissues. The cisplatin levels did not show marked change in lung, while it significantly decreased in the heart and brain 4 days after cisplatin injection compared to the initial day. The results strongly suggest that cisplatin has an inhibitory effect on ATP synthase. The levels of ATP displayed positive correlation between lung and brain of cisplatin-treated rats during all periods (*r* = 0.956). Alterations in the ADP levels may be explained by nucleotide synthesis being affected in the lung and heart compared to the brain.

Capsaicin is an important plant compound and a major pungent ingredient of hot pepper and is used in food, as spices, and in medicines and it is known as non-enzymatic antioxidant and it is the major capsaicinoid present in peppers [47,48,49,50]. The present study aimed to determine the possible protective effect of capsaicin against toxicity induced by cisplatin in lung, heart, and brain tissues of rats. In the rats injected with cisplatin + capsaicin, the SDH activities and the LPO levels were higher and lower in heart and brain tissues than in the cisplatin group, respectively. The results indicated that capsaicin has remarkable radical scavenging capacity at/near the membrane surface and in the interior of the phospholipid membrane [51]. This corresponds with the results of Lee et al., who found that rats treated with capsaicin showed a reduction in oxidative stress measured as MDA in the liver, kidney, and muscle [52]. However, in the cisplatin + capsaicin groups, there was an observation of decreases in both the SDH and COX activities in the lung tissues in comparison with control and cisplatin groups. Lung was likely impaired by capsaicin besides cisplatin because capsaicin probably induced toxicity by irritation and sensitization in lung tissues. In addition, the determination of lower LPO levels as well as increases in CAT activities in lung cisplatin + capsaicin group compared to both control and cisplatin groups may strongly show that CAT enzyme has important oxygen radical scavenging capacities. Cisplatin promotes apoptosis by ROS production, but it also causes inflammation and increased mitochondrial damage. Therefore, using antioxidants such as curcumin, ellagic acid, and lipoic acid to prevent this pathology could be advisable [17]. As discussed above, capsaicin is one such antioxidant. Capsaicin has the property of scavenging ROS such as hydroxyl and alkoxy radicals, so it reduces lipid peroxidation and protein carbonyl formation [17]. In the literature, it has been shown that the effect of capsaicin protection is against ROS and lipid peroxidation which was enhanced by cisplatin [53]. On the other hand, capsaicin antioxidant properties can affect the therapeutic mechanism of the drugs which induce ROS production as a therapeutic effect. It is worth noting that the protective effects of antioxidants against drug toxicity are very important for the survival of healthy tissues during the treatment process. The ATP and ADP levels in all studied tissues of cisplatin + capsaicin exposed rats were higher compared to control except, for ADP levels of heart. In this study, these results indicated that capsaicin reduced cisplatin toxicity. Additionally, this toxicity’s attenuation may be explained by the protection of ATP synthesis enzymes which are inhibited by cisplatin toxicity.

## 5. Conclusions

The present research displayed that cisplatin induced toxicity due to the impairment of the balance between ETC and the antioxidant system. Rats were also injected with both capsaicin and cisplatin to evaluate the antioxidative and protective impact of the capsaicin on the studied systems involved in ETC, LPO, and CAT. The results of the cisplatin + capsaicin treatment group suggest the capacity of the antioxidant of capsaicin to scavenge ROS and prevent membrane destruction.

## Figures and Tables

**Figure 1 molecules-23-03267-f001:**
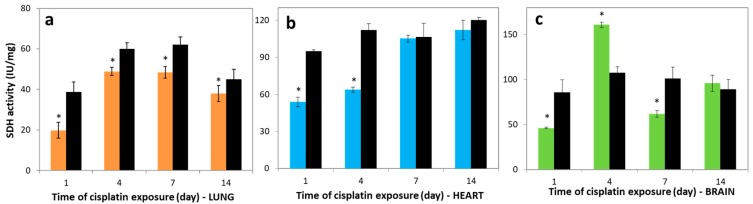
Succinate dehydrogenase (SDH) activity in (**a**) lung, (**b**) heart, and (**c**) brain tissues over time. Lung (―■―), heart (―■―), and brain (―■―) treatment groups, and controls (―■―). The results are expressed as mean ± SD (*n* = 6). * Significantly different (*p* < 0.05) from control.

**Figure 2 molecules-23-03267-f002:**
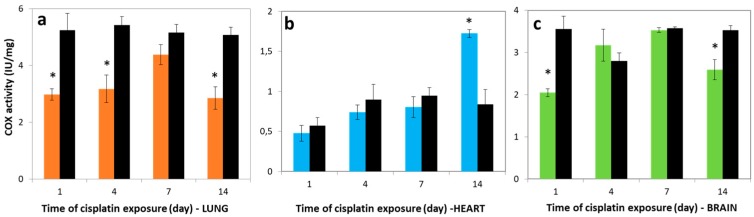
Cytochrome c oxidase (COX) activity in (**a**) lung, (**b**) heart, and (**c**) brain tissues over time. Lung (―■―), heart (―■―), and brain (―■―) treatment groups, and controls (―■―). Results are expressed as mean ± SD (*n* = 6). * Significantly different (*p* < 0.05) from control.

**Figure 3 molecules-23-03267-f003:**
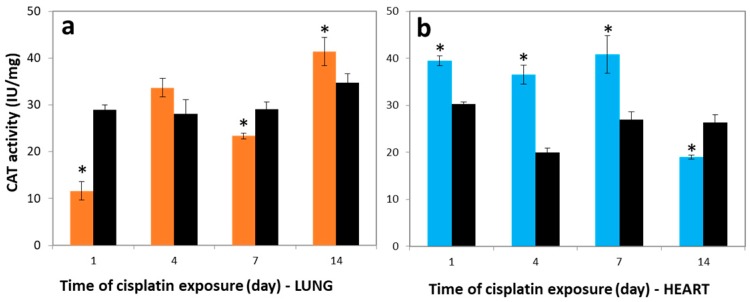
Catalase (CAT) activity in (**a**) lung and (**b**) heart tissues over time. Lung (―■―) and heart (―■―) tissues of treatment group, and controls (―■―). These results are expressed as mean ± SD (*n* = 6). * Significantly different (*p* < 0.05) from control.

**Figure 4 molecules-23-03267-f004:**
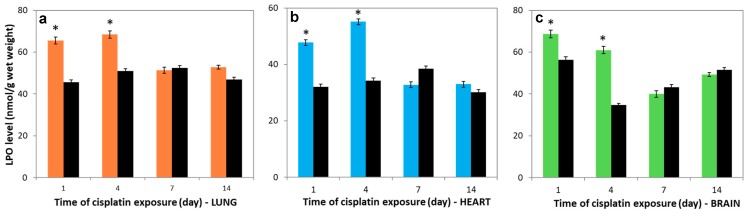
Lipid peroxidation (LPO) levels in (**a**) lung, (**b**) heart and (**c**) brain tissues over time. Lung (―■―), heart (―■―), and brain (―■―) tissues of treatment group, and controls (―■―). Results are expressed as mean ± SD (*n* = 6). * Significantly different (*p* < 0.05) from control.

**Figure 5 molecules-23-03267-f005:**
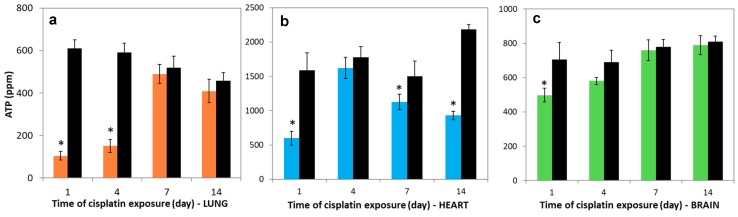
ATP levels in (**a**) lung, (**b**) heart, and (**c**) brain tissues over time. Lung (―■―), heart (―■―), and brain (―■―) tissue of treatment group, and controls (―■―). Results are expressed as mean ± SD (*n* = 6). * Significantly different (*p* < 0.05) from control.

**Figure 6 molecules-23-03267-f006:**
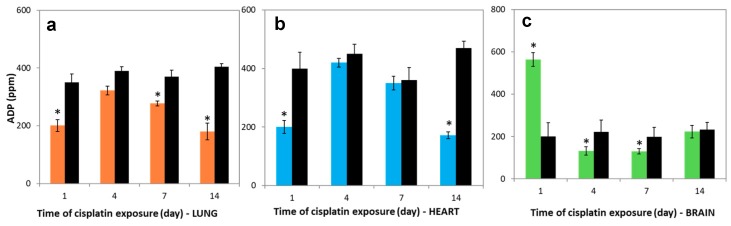
ADP levels in (**a**) lung, (**b**) heart, and (**c**) brain tissues over time. Lung (―■―), heart (―■―), and brain (―■―) tissues of treatment group, and controls (―■―). Results are expressed as mean ± SD (*n* = 6). * Significantly different (*p* < 0.05) from control.

**Figure 7 molecules-23-03267-f007:**
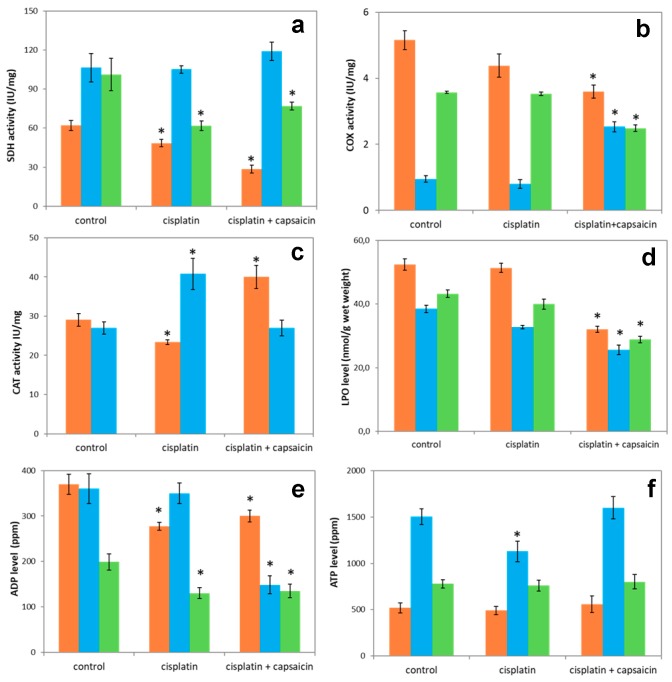
(**a**) SDH activity, (**b**) COX activity, (**c**) CAT activity, (**d**) LPO levels, (**e**) ADP levels, and (**f**) ATP levels in Lung (―■―), heart (―■―), and brain (―■―) tissues for the control and cisplatin groups for the 7th day, and group with capsaicin exposure followed by 7 days of cisplatin injection. These results are expressed as mean ± SD (*n* = 6). * Significantly different (*p* < 0.05) from control.

**Table 1 molecules-23-03267-t001:** Platin levels in 1 g of lung, heart and brain tissues.

	Lung	Heart	Brain
**1st Day (ppm platin)**	0.24	0.15	0.24
**4th Day (ppm platin)**	0.23	0.039	Non Detectable

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
