# Peer review of "The Investigation of the Antitumor Agent Toxicity and Capsaicin Effect on the Electron Transport Chain Enzymes, Catalase Activities and Lipid Peroxidation Levels in Lung, Heart and Brain Tissues of Rats"

_molecules, 2018, doi:10.3390/molecules23123267_

Round 1

Reviewer 1 Report

It is difficult to perceive a large number of similar diagrams in the text of the article. I would like to see the absolute numerical values, and clarify the level of significance (p) to confirm the identified significant differences. When mentioning correlations between parameters in the test, it would also be appropriate to indicate the values of the correlation coefficients. In general, it is necessary to justify the use of parametric statistics for processing results. Has the distribution of the experimental data been verified?

The discussion did not focus on the practical application of the results obtained. Not limited to the study. Do antioxidants affect the therapeutic effect of the drug? Does the antioxidant mechanism of action matter? What will be the development of research? In my opinion, this part of the research is the most interesting and practically significant.

Author Response

·        It is difficult to perceive a large number of similar diagrams in the text of the article.

     Many thanks for the advice, the similarities have been rearranged.

·        I would like to see the absolute numerical values, and clarify the level of significance (p) to confirm the identified significant differences.

     Thank you very much for your warning, absolute numerical values have been written in the sentences and clarifications of significance (p).

·        When mentioning correlations between parameters in the test, it would also be appropriate to indicate the values of the correlation coefficients.

     Thank you very much for your warning, most of the correlation coefficients in this study has been organized.

·        In general, it is necessary to justify the use of parametric statistics for processing results. Has the distribution of the experimental data been verified?

     The Shapiro–Wilk test has been used to test the normality assumption.

·        The discussion did not focus on the practical application of the results obtained. Not limited to the study.

     Thank you for your advice, discussion part has been reconstructed.  

·        Do antioxidants affect the therapeutic effect of the drug? Does the antioxidant mechanism of action matter?

      Thank you for reminding and contribution. It has been explained in the discussion part as “Cisplatin promotes apoptosis by ROS production but it causes also inflammation, increased mitochondrial damage, therefore, using antioxidants such as curcumin, ellagic acid, lipoic acid to prevent this pathology could be advised [17]. One of these antioxidants is capsaicin which is the major capsaicinoid present in peppers [49-52]. Capsaicin has the property of scavenging ROS, such as hydroxyl and alkoxy radicals so it reduces lipid peroxidation and protein carbonyl formation [17]. In literature, it is shown that the effect of capsaicin protection is against ROS and lipid peroxidation which was enhanced by cisplatin [53]. On the other hand, capsaicin antioxidant properties can affect the therapeutic mechanism of the drugs which induces ROS production as a therapeutic effect. By the way, protective effects of antioxidants against drug toxicity is very important to survive in treatment process for healthy tissues.”

·        What will be the development of research? In my opinion, this part of the research is the most interesting and practically significant.

     In this study, we have shown that capsaicin as an antioxidant reduced the damage of cisplatin treatment. This shows the importance of using antioxidants in chemotherapeutic therapy. The next step of this study, could lead to new research and give direction to new studies about decreasing drug toxicity which is the most important problem in cancer treatment. 

Reviewer 2 Report

REVIEWER COMMENTS

The paper deals with the effect of cisplatin on mitochondrial function of different tissues of rats, and the effect of capsaicin, as antioxidant, against free radical-initiated damages.

The experiment methodically is almost correct, but description of the treatment is not clear, and there is no information about how the dose of cisplatin and capsaicin were calculated, and capsaicin dissolved in serum, due to unknown reason, but it may arise local immune response, therefore not acceptable.

Some statements about free radical formation and the antioxidant defence are not correct. It is problematic in the discussion, which requires corrections, in particular about the possible effect of capsaicin, which is missing.

Otherwise, there is some critical format problem with the figures.

Comments in order of appearance in the text:

r. 43-44: please change the order: (1) superoxide, (2) peroxyl radical, (3) hydrogen peroxide. Peroxinitrite is not ROS but RNS

r. 56.  antioxidant enzymes are not scavengers, those neutralise ROS/RNS but not scavenge

r. 57. please use GPx instead of GSH-Px

r. 58. vitamin E and glutathione are scavengers, but not catalase

r. 63. Which parameters investigated? It is an unclear sentence

r. 79. ppm/gr – it is a false dimension, please modify

Table 1 - ppm/gr – it is a false dimension, please modify

Figures 1-7 please add marks for significant differences not only in the text but also in Figures. This is a critical format error.

r. 123-124 This sentence is incorrect. CAT breakdown hydrogen peroxide, but not as effect of electron leakage, because it results superoxide release, which at first should dismutate to hydrogen peroxide by SOD

r. 131-132. low CAT activity on peroxisomes – it is nonsense

r. 238. Hydrogen peroxide never produced by ETC enzymes. Those may produce superoxide anion

r. 239-240 It is nonsence (again)

r. 306-309. Description is unclear, when and which treatment applied. Method of calculation of the dose of cisplatin and capsaicin also missing.  If you dissolved capsaicin in serum and injected intramuscularly it may arise local immune response, therefore this method is not acceptable.

r. 440-441. This reference about low catalase activity in peroxisome, which, as the reviewer knows is nonsense, but it was not possible to check the cited paper. Please add another reference which published in English and available in peer-reviewed periodical, because the present one is not acceptable, because it is not available for the international scientific community

r. 457-458. typing error!

Author Response

The paper deals with the effect of cisplatin on mitochondrial function of different tissues of rats, and the effect of capsaicin, as antioxidant, against free radical-initiated damages.

The experiment methodically is almost correct, but description of the treatment is not clear, and there is no information about how the dose of cisplatin and capsaicin were calculated, and capsaicin dissolved in serum, due to unknown reason, but it may arise local immune response, therefore not acceptable.

     Capsaicin has been dissolved in ethanol before serum. Capsaicin is dissolved in organic solvents such as ethanol, DMSO and dimethyl formamide according to product information paper. ethanol as a solvent for dissolving of capsaicin has been selected and then solved into serum for injection. We referred the publication for the concentration for cisplatin and capsaicin in material and method part, indeed these doses were treated after literature research. Injection of capsaicin and cisplatin doses were chosen depending on the previous literatures (Yuka et al. and Mashhadi et al.) For control groups, serum physiologic has been injected to rats at 1st day.

Some statements about free radical formation and the antioxidant defence are not correct. It is problematic in the discussion, which requires corrections, in particular about the possible effect of capsaicin, which is missing.

Otherwise, there is some critical format problem with the figures.

Comments in order of appearance in the text:

     Thank you for your valuable comments, corrections and advices.

r. 43-44: please change the order: (1) superoxide, (2) peroxyl radical, (3) hydrogen peroxide. Peroxinitrite is not ROS but RNS

     The order has been carried out in specifically.

r. 56.  antioxidant enzymes are not scavengers, those neutralise ROS/RNS but not scavenge

     Well reorganized.

r. 57. please use GPx instead of GSH-Px

     This abbreviation has been changed.

r. 58. vitamin E and glutathione are scavengers, but not catalase

     Well reorganized.

r. 63. Which parameters investigated? It is an unclear sentence

     Mentioned parameters which are SDH, COX and CAT enzyme activities, LPO and nucleotides levels of lung, heart and brain tissues induced by cisplatin have been explained.

r. 79. ppm/gr – it is a false dimension, please modify

Table 1 - ppm/gr – it is a false dimension, please modify

     The dimension of “ppm/gr” was changed to “0.24, 0.24, 0.15 ppm cisplatin in the 1 gr tissue”.

Figures 1-7 please add marks for significant differences not only in the text but also in Figures. This is a critical format error.

     Significant differences were added in all Figures.

r. 123-124 This sentence is incorrect. CAT breakdown hydrogen peroxide, but not as effect of electron leakage, because it results superoxide release, which at first should dismutate to hydrogen peroxide by SOD.

     This sentence has been restructured. It mentioned that firstly, superoxide is produced electron leakage in ETC and then secondly, superoxide is reduced to hydrogen peroxide by catalase enzyme.

r. 131-132. low CAT activity on peroxisomes – it is nonsense

r. 238. Hydrogen peroxide never produced by ETC enzymes. Those may produce superoxide anion

r. 239-240 It is nonsense (again)

r. 440-441. This reference about low catalase activity in peroxisome, which, as the reviewer knows is nonsense, but it was not possible to check the cited paper. Please add another reference which published in English and available in peer-reviewed periodical, because the present one is not acceptable, because it is not available for the international scientific community

     Thank you for corrections about r. 131-132, r. 239-240, r. 440-441 in the discussion part and r. 238 in the result part have been well reconstructed

r. 306-309. Description is unclear, when and which treatment applied. Method of calculation of the dose of cisplatin and capsaicin also missing.  If you dissolved capsaicin in serum and injected intramuscularly it may arise local immune response, therefore this method is not acceptable.

     These explanations and references have been added in the material method part.

r. 457-458. typing error!

     It was corrected.

Reviewer 3 Report

Manuscript is interesting and well ellaborated. It could be useful for starts new research in the area, and also for students.

As minor comments:

* Please improve the Introduction part, giving more details and information.

* Please proviode more information for the SPSS version used, as Company (city, country), or (city, state, USA).

Author Response

Manuscript is interesting and well elaborated. It could be useful for starts new research in the area, and also for students.

As minor comments:

·      Please improve the Introduction part, giving more details and information.

     Thank you for advice, well reorganized.

·      Please provide more information for the SPSS version used, as Company (city, country), or (city, state, USA)

    Thank you for your correction, this information has been added to material method part as “SPSS 15.0 Windows, US Government, 233 South Wacker Drive, 11th Floor Chicago”.

Round 2

Reviewer 2 Report

The revised version is improved the quality of the paper and the Authors corrected all of my proposals, therefore it is acceptable for publication.